# Brain Glucose Hypometabolism and Iron Accumulation in Different Brain Regions in Alzheimer’s and Parkinson’s Diseases

**DOI:** 10.3390/ph15050551

**Published:** 2022-04-29

**Authors:** Indira Y. Rao, Leah R. Hanson, Julia C. Johnson, Michael H. Rosenbloom, William H. Frey

**Affiliations:** 1HealthPartners Center for Memory and Aging, 295 Phalen Boulevard, St. Paul, MN 55130, USA; rao00111@umn.edu (I.Y.R.); leah.r.hanson@healthpartners.com (L.R.H.); michael.h.rosenbloom@healthpartners.com (M.H.R.); 2HealthPartners Institute, Bloomington, MN 55425, USA; 3HealthPartners Struthers Parkinson’s Center, Minneapolis, MN 55427, USA; julia.c.johnson2@healthpartners.com

**Keywords:** glucose hypometabolism, brain iron accumulation, FDG-PET, QSM–MRI, intranasal insulin, intranasal deferoxamine, Alzheimer’s disease, Parkinson’s disease

## Abstract

The aim of this study was to examine the relationship between the presence of glucose hypometabolism (GHM) and brain iron accumulation (BIA), two potential pathological mechanisms in neurodegenerative disease, in different regions of the brain in people with late-onset Alzheimer’s disease (AD) or Parkinson’s disease (PD). Studies that conducted fluorodeoxyglucose positron emission tomography (FDG-PET) to map GHM or quantitative susceptibility mapping—magnetic resonance imaging (QSM–MRI) to map BIA in the brains of patients with AD or PD were reviewed. Regions of the brain where GHM or BIA were reported in each disease were compared. In AD, both GHM and BIA were reported in the hippocampus, temporal, and parietal lobes. GHM alone was reported in the cingulate gyrus, precuneus and occipital lobe. BIA alone was reported in the caudate nucleus, putamen and globus pallidus. In PD, both GHM and BIA were reported in thalamus, globus pallidus, putamen, hippocampus, and temporal and frontal lobes. GHM alone was reported in cingulate gyrus, caudate nucleus, cerebellum, and parietal and occipital lobes. BIA alone was reported in the substantia nigra and red nucleus. GHM and BIA are observed independent of one another in various brain regions in both AD and PD. This suggests that GHM is not always necessary or sufficient to cause BIA and vice versa. Hypothesis-driven FDG-PET and QSM–MRI imaging studies, where both are conducted on individuals with AD or PD, are needed to confirm or disprove the observations presented here about the potential relationship or lack thereof between GHM and BIA in AD and PD.

## 1. Introduction

In neurodegenerative diseases such as AD and PD, biomarkers have been recently employed in clinical practice and research to help confirm the disease process, evaluate progression, and indicate potential targets for therapeutic interventions. In AD, amyloid plaques and neurofibrillary tangles characterize the disease from a neuropathological standpoint, although alternative theories exist regarding whether these factors or others, such as insulin signaling abnormalities, account for disease progression. PD is pathologically characterized as an alpha-synucleinopathy, which starts within the olfactory bulbs or myenteric plexus, eventually spreading to the brainstem and subcortical regions over time. Biomarkers may be disease specific such as cerebrospinal beta amyloid and phosphorylated tau in AD or may be non-specific, offering evidence for neurodegeneration in the form of decreased cerebral glucose uptake and metabolism, and the abnormal accumulation of iron. This review focuses on two neurodegenerative biomarkers—glucose hypometabolism (GHM) and excessive brain iron accumulation (BIA) in AD and PD.

Brain fluorodeoxyglucose positron emission tomography (FDG-PET) imaging studies have demonstrated that glucose uptake and utilization in the brains of individuals with AD and PD are greatly reduced [1,2]. Herholz et al., stated “Many studies over the past two decades have demonstrated that AD is associated with regional reductions of the cerebral metabolic rate of glucose (CMRGlc) mainly in precuneus, temporo-parietal, and frontal association cortex.” [3]. This GHM is consistent with much earlier biochemical studies on samples of temporal lobe obtained at autopsy from patients with AD who demonstrated a marked decrease in glucose metabolism when compared to tissue from age-matched non-demented control subjects [4]. Since the primary form of energy for neurons is glucose, both normal brain function and the processes of repairing and replacing cellular components essential for brain function when they are lost or damaged due to aging and disease are likely impaired by greatly reduced glucose metabolism. As early as 2008, Mosconi et al. noted that FDG-PET studies demonstrated GHM in key brain regions accurately distinguishes AD from normal aging and occurs before cognitive decline and the onset of dementia [1]. GHM is also found in cognitively normal subjects with subjective memory complaints and in carriers of the APOE E4 genotype [5]. Cortical GHM reflects local atrophy and tau pathology in symptomatic patients with AD [6]. Meles et al. [7] describe the PD-related pattern of altered glucose metabolism, including relative hypometabolism in the temporal, parietal, occipital, and frontal cortices, which have been reported to correlate significantly with motor symptoms and presynaptic dopaminergic deficits in the posterior striatum. Anandhan et al. [8] reviewed evidence that also supports the contribution of altered glucose metabolism to dopaminergic cell death.

Brain iron abnormally accumulates in individuals with AD, PD, and a number of other neurological disorders, which leads to free radical generation, oxidative damage and inactivation of key brain components [9]. Free heme has been reported to increase approximately 2.5-fold in the brains of individuals with AD [10,11]. Gao et al., reported elevated iron in the brains of AD patients as compared to normal controls in the bilateral hippocampus, dentate nucleus, red nucleus, putamen, and lateral temporal cortex [12]. The impaired redox balance of the iron species Fe^2+^ vs. Fe^3+^ contributes to oxidative stress [9]. Defective homeostasis of iron probably contributes to neuropathology, namely protein misfolding of A-beta and phospho-tau in AD and iron co-localizes with amyloid plaques and neurofibrillary tangles [9]. In addition, the human brain muscarinic cholinergic receptor required for memory has been shown in vitro to be inactivated by both free iron and heme iron [13]. Finally, MRI brain scans of post-mortem human brains show decreases in hippocampal T2*, which is sensitive to changes in the magnetic properties of iron [14,15]. Postmortem studies have demonstrated increased total iron and iron (III) in the substantia nigra of Parkinson’s patients compared to age-matched controls [16], and iron deposition in neurons has been proposed to contribute to the development of alpha-synuclein pathology in PD [17]. A 2016 meta-analysis revealed that other than the substantia nigra, iron deposits have also been observed in brain regions such as the red nuclei, globus pallidus, and cortex of PD patients [18]. Consistent with the above, BIA has been linked with cognitive severity in PD [19].

Potential treatments to target both GHM and BIA biomarkers have been developed and tested to improve the treatment of AD and PD. Insulin resistance and decreased insulin signaling in the brain have been considered by many to contribute to both GHM, decreased brain cell energy and cognitive decline in AD [20,21]. Development of the intranasal insulin treatment [22] has led to multiple human clinical trials demonstrating increased brain cell energy in healthy adults [23] and improved memory in healthy adults [24] and individuals with mild cognitive impairment or AD [25,26]. A pilot clinical trial of intranasal insulin in individuals with PD also showed promising results with indication of clinically relevant functional improvement [27]. There is a strong rationale and extensive preclinical support in animal models of AD, PD and stroke for reducing BIA with intranasal deferoxamine, a high-affinity iron chelator, as a treatment for neurodegenerative and neurovascular disorders [28]. The potential of these treatments makes it important to understand the relationship between GHM and BIA. Studies that assessed brain iron in individuals with AD or PD using quantitative susceptibility mapping—magnetic resonance imaging (QSM–MRI) or GHM using FDG-PET are reviewed herein. The examination of the relationship of the presence of GHM and BIA in brain regions of individuals with AD and PD could reveal important aspects in their association, which may be beneficial in developing not only a better understanding of the pathophysiology of AD and PD, but may also inform future treatments addressing GHM and BIA in these neurodegenerative diseases.

## 2. Methods

To describe the anatomical relationship between the distribution of GHM and BIA in different brain regions, published literature on GHM and BIA in AD and PD was identified through PubMed using the search terms glucose hypometabolism, brain iron accumulation, Alzheimer’s disease, Parkinson’s disease, FDG-PET, and QSM–MRI. Studies that conducted FDG-PET scans to locate regions with elevated GHM in the brains of patients with AD or PD as compared to healthy controls were examined. In addition, studies that conducted QSM–MRI on the brains of patients with AD or PD as compared to healthy control brains were selected and regions where these studies reported elevated BIA were examined.

Studies were removed from consideration for several reasons. Studies that recruited subjects with young/early onset AD or PD were not included since these cases likely involve different causative factors, such as gene mutations [29,30]. Additionally, papers that examined only two or fewer brain regions were not included due to the number of regions of interest being insufficient for a whole brain analysis. Papers published prior to the year 2000 were excluded due to changes in technology and methodology of brain imaging and data analysis. In the case of AD, studies that only compared patients with mild cognitive impairment (MCI) to healthy controls were excluded. In PD, only studies that examined patients with idiopathic Parkinson’s disease were reviewed, while studies on individuals with other Parkinsonian diseases, dementia with Lewy Bodies, etc., were not included. Postmortem studies were excluded.

In summarizing the presence of GHM and BIA in brain regions, there were several guiding principles. If two or more papers reported finding either of the two biomarkers in a certain region, the region was recorded and included in the summary. When there was a difference in specificity of brain regions where a particular biomarker was seen, both whole lobes and their more specific parts were included in the table. Because the hippocampus is part of the temporal lobe, it is also included in the temporal lobe column of the tables, even if the cited paper only mentioned the hippocampus. When only one paper mentioned a very specific region of a lobe, for example, the parahippocampal gyrus, the temporal lobe was marked with a “+”, but the parahippocampal gyrus was not added as a region to the table.

## 3. Results

In AD, 10 studies were reviewed for GHM and 8 studies for BIA. However, not all studies examined all regions. GHM was examined and reported present in cingulate gyrus, temporal lobe, parietal lobe, precuneus, hippocampus, frontal lobe, and occipital lobe (Table 1a). BIA was examined and reported present in the hippocampus, caudate nucleus, putamen, globus pallidus, temporal, and parietal lobes (Table 1b). All 10 of the studies that examined the temporal lobe reported GHM present, while all 5 studies that examined this region for BIA reported it present. All nine studies that examined the cingulate gyrus and the parietal lobe for GHM reported it present, while only one study examined and reported BIA present in cingulate gyrus and two studies examined and reported BIA present in parietal lobe. Six of the seven studies that examined BIA in the caudate nucleus reported it present, while only one of the two studies that examined GHM there reported it present. All six of the studies that examined BIA in the putamen reported it present, while the one study that examined GHM in the putamen reported it absent. Only GHM was reported present in the precuneus and occipital lobe, while only BIA was reported present in the putamen and globus pallidus. GHM and BIA were both reported present in the temporal and parietal lobes. Both biomarkers were also observed in the hippocampus. All three of the studies that examined GHM and all five of the studies that examined BIA reported it present in the hippocampus.

In PD, 15 studies were reviewed for GHM, and 15 studies were reviewed for BIA. GHM was reported present in parietal, occipital, frontal, and temporal lobes, the cerebellum, cingulate gyrus, thalamus, caudate nucleus, globus pallidus, putamen, and hippocampus (Table 2a). BIA was reported present in the substantia nigra, red nucleus, globus pallidus, putamen, thalamus, hippocampus, and frontal and temporal lobes (Table 2b). All of the reviewed papers reported GHM present in the parietal lobe. GHM was also reported present in 12 of the 14 studies that examined the occipital lobe, 11 of the 14 studies that examined the frontal lobe, and 10 of the 11 studies that examined the temporal lobe. Only one study examined and reported BIA present in the parietal lobe, and only two examined and reported it present in the frontal and temporal lobes. The one study that examined the occipital lobe reported it absent. Thirteen of the fifteen studies that examined BIA in the substantia nigra reported it present, and six of the fourteen studies that examined it in the red nucleus reported it present. None of the papers examined or reported GHM in these two regions.

Comparing the relationship of the presence of GHM and BIA between AD and PD reveals that GHM is present in the parietal and temporal lobes in more than 90% of the reviewed studies in both diseases (Table 3). In AD, while all of the studies that examined the cingulate gyrus and precuneus reported the presence of GHM in these regions, only one of the studies examined and reported BIA present in the cingulate gyrus and none examined BIA in the precuneus. In PD, while BIA was reported present in the substantia nigra in 87% of the PD studies, no study examined GHM in this region. The anatomical distribution of GHM and BIA in AD and PD is graphically displayed in Figure 1.

## 4. Discussion

This is the first study to explicitly describe the spatial relationship in the brain between GHM and BIA, two biomarkers that are elevated in comparison to healthy controls in both AD and PD, by comparing data from brain FDG-PET and QSM–MRI studies. In AD, GHM and BIA were both reported in the hippocampus, and temporal and parietal lobes. It is likely that regions of the brain where both GHM and BIA occur will experience more damage than those showing only one of the two. For example, the hippocampus, which shows both GHM and BIA, is widely accepted as an important memory region whose functioning is greatly reduced in AD. Further, a 2017 study found that the medial temporal lobe, which includes the hippocampus, shows significant atrophy in AD, adding to the evidence [76]. Neurofibrillary tangles appear earlier in the disease process [77] and have also been observed with tau-PET in this region [78]. Since beta amyloid binds both iron [79] and free heme [11], it is not surprising that many of the regions typically associated with AD are those that show both beta amyloid, and consequently, BIA [11,80].

In PD however, one of the most affected motor regions, the substantia nigra, was only reported to show BIA. However other regions such as the globus pallidus, putamen, and thalamus that are also affected in Parkinson’s disease were reported to show both GHM and BIA. Further, in PD, the region that was reported by most studies to be affected by GHM was the parietal lobe, a region involved in coordinating movement. However, this region does not accumulate alpha synuclein and Lewy bodies to the degree that the substantia nigra does, and its involvement in PD is not always appreciated. However, connecting this finding to recent literature on the human connectome model of brain function aids in understanding this finding. This model suggests that rather than individual parts of the brain performing specific functions, networks that span across different lobes of the brain are responsible for functions [81]. According to this model, regions such as the parietal lobe would be part of a complex network of neurons associated with movement. Thus, when these regions are starved for energy and show GHM in PD, patients suffer from motor symptoms. This study suggests that although the parietal lobe is not a region that typically contains Lewy bodies, it is also an important region to target in PD [82].

Several of the studies, included in the tables above, selected regions of interest based on regions where previous literature had reported the presence of GHM or BIA. If multiple previous studies did not report significant GHM or BIA in a certain brain region, subsequent studies would be unlikely to report on that brain region unless they observed significant GHM or BIA there. The results suggest that GHM and BIA exist independent of each other in several brain regions, including the putamen, occipital lobe, precuneus, and globus pallidus in AD and substantia nigra, red nucleus, cingulate gyrus, occipital lobe, and cerebellum in PD. However, hypothesis-driven FDG-PET and QSM–MRI imaging studies, where both are conducted in the same brain regions on each group of individuals with AD or PD, will be needed to confirm or disprove the tentative observations presented here about the potential relationship or lack thereof between GHM and BIA in AD and PD.

Our analysis was limited by the number of published studies on GHM and BIA in individuals with AD or PD. Furthermore, the published studies did not always identify the same regions of interest and did not consider the medications that the patients were taking at the time the FDG-PET or QSM–MRI brain imaging studies were performed. For example, subjects with levodopa-induced dyskinesia have more iron in the substantia nigra than those without it [83]. It is also apparent that the results would be more accurate if studies were identified where FDG-PET and QSM–MRI were both conducted on the same patients or on age- and sex-matched pairs. However, no such studies exist.

Brain biomarkers representing potential pathologic mechanisms, such as GHM and BIA, have also been considered as therapeutic targets. Multiple preclinical studies on intranasal deferoxamine, a high-affinity iron chelator, have demonstrated improvements in rodent models of AD and PD [28]. Intranasal deferoxamine has even been reported to improve memory from baseline, increase levels of HIF-1α, and inhibit GSK-3β activity in healthy C57 mice. A much older study reported that intramuscular administration of deferoxamine reduced cognitive decline in AD patients by 50% over a period of two years [84]. Intranasal insulin, a treatment targeting GHM, has been reported to be both safe and beneficial in humans. Multiple human clinical trials have demonstrated that intranasal insulin improves memory in normal healthy adults with no change in the blood levels of insulin or glucose [20,24,26]. In patients with mild cognitive impairment (MCI) or AD, a single dose of intranasal insulin improves memory [25], and intranasal insulin also improves memory with multiple treatments of patients with AD or MCI [20]. A pilot clinical trial on intranasal insulin in individuals with PD also showed promising results with indication of clinically relevant functional improvement including improved verbal fluency and movement, and decreased physical disability [27]. Therefore, it is becoming increasingly more important to learn more about these biomarkers. Since GHM and BIA often occur in different brain regions as shown here, it may be clinically more effective to treat AD and PD by administering a combination of drugs targeting both GHM and BIA.

## 5. Conclusions

The results suggest that GHM and BIA are observed independent of one another in various brain regions in both AD and PD. This suggests that GHM is not always necessary or sufficient to cause BIA in a brain region and vice versa. In AD, the brain regions most often reported to show both GHM and BIA in published studies are the temporal lobe and more specifically, the hippocampus, which is key to memory function and demonstrates marked degeneration and volume loss in patients with AD. In PD, while GHM was most often reported in the parietal, occipital, frontal, and temporal lobes, BIA was most often reported in the substantia nigra, red nucleus, and globus pallidus. Hypothesis-driven FDG-PET and QSM–MRI imaging studies, where both are conducted on each group of individuals with AD or PD, are needed to confirm or disprove the tentative observations presented here about the potential relationship or lack thereof between GHM and BIA in AD and PD.

## Figures and Tables

**Figure 1 pharmaceuticals-15-00551-f001:**
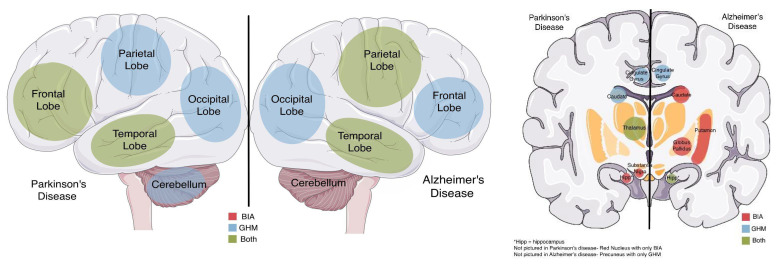
The predominant distribution of brain iron accumulation (BIA), glucose hypometabolism (GHM), or both for Parkinson’s disease and Alzheimer’s disease, based on Table 3, is presented. (This schematic drawing was created using art elements from Servier Medical Art Commons Attribution 3.0 Unported License. Servier Medical Art by Servier is licensed under a Creative Commons Attribution 3.0 Unported License).

**Table 1 pharmaceuticals-15-00551-t001:** Glucose Hypometabolism and Brain Iron Accumulation in Alzheimer’s Disease.

**a.** Glucose hypometabolism (GHM) in AD: present (+), absent (−), not reported (NR). Papers reporting parieto–temporal location are shown by merging the columns for the parietal and temporal lobes. Under each brain region is the number of studies reporting GHM present over the number of studies that examined this region.										
**Study Number**	**Temporal Lobe**	**Parietal** **Lobe**	**Cingulate Gyrus**	**Precuneus**	**Hippocampus**	**Frontal** **Lobe**	**Occipital Lobe**	**Caudate Nucleus**	**Putamen**	**Globus Pallidus**
	10/10	9/9	9/9	5/5	3/3	3/3	2/3	1/2	0/1	0/0
[3]	**+**		**+**	**+**	NR	**+**	NR	−	−	NR
[31]	**+**	NR	**+**	NR	NR	NR	NR	NR	NR	NR
[32]	**+**	**+**	**+**	**+**	NR	NR	NR	NR	NR	NR
[33]	**+**	**+**	**+**	**+**	**+**	NR	NR	NR	NR	NR
[34]	**+**		**+**	NR	NR	NR	NR	NR	NR	NR
[35]	**+**		**+**	NR	**+**	NR	−	NR	NR	NR
[36]	**+**	**+**	**+**	**+**	**+**	NR	NR	**+**	NR	NR
[37]	**+**	**+**	NR	NR	NR	**+**	**+**	NR	NR	NR
[38]	**+**	**+**	**+**	NR	NR	**+**	**+**	NR	NR	NR
[39]	**+**		**+**	**+**	NR	NR	NR	NR	NR	NR
**b.** Brain iron accumulation (BIA) in AD: present (+), absent (–), not reported (NR). Under each brain region is the number of studies reporting BIA present over the number of studies that examined this region.										
**Study Number**	**Temporal Lobe**	**Parietal Lobe**	**Cingulate Gyrus**	**Precuneus**	**Hippocampus**	**Frontal Lobe**	**Occipital Lobe**	**Caudate Nucleus**	**Putamen**	**Globus Pallidus**
	5/5	2/2	1/1	0/0	5/5	1/3	0/0	6/7	6/6	2/6
[40]	**+**	NR	NR	NR	**+**	NR	NR	**+**	**+**	NR
[41]	**+**	NR	NR	NR	**+**	NR	NR	NR	NR	NR
[42]	NR	NR	NR	NR	NR	−	NR	**+**	**+**	**+**
[43]	NR	NR	NR	NR	NR	NR	NR	**+**	**+**	−
[44]	NR	NR	NR	NR	NR	NR	NR	**+**	**+**	−
[45]	**+**	NR	NR	NR	**+**	NR	NR	−	**+**	**+**
[46]	**+**	**+**	NR	NR	**+**	−	NR	**+**	**+**	−
[47]	**+**	**+**	**+**	NR	**+**	**+**	NR	**+**	NR	−

**Table 2 pharmaceuticals-15-00551-t002:** Glucose Hypometabolism and Brain Iron Accumulation in Parkinson’s Disease.

**a.** Glucose hypometabolism (GHM) in PD: present (+), absent (−), not reported (NR), Parkinson’s disease dementia (PDD). Under each brain region is the number of studies reporting GHM present over the number of studies that examined this region.													
**Study Number**	**Substantia Nigra**	**Red Nucleus**	**Parietal** **Lobe**	**Occipital Lobe**	**Frontal Lobe**	**Temporal Lobe**	**Cerebellum**	**Cingulate** **Gyrus**	**Thalamus**	**Caudate Nucleus**	**Putamen**	**Globus** **Pallidus**	**Hippocampus**
	0/0	0/0	15/15	11/14	11/14	10/11	6/10	6/8	5/8	5/6	2/6	2/4	2/3
[48]	NR	NR	**+**	**+**	−	−	−	NR	−	NR	−	−	NR
[7]	NR	NR	**+**	**+**	**+**	NR	**+**	NR	**+**	NR	**+**	**+**	NR
[49]	NR	NR	**+**	**+**	**+**	**+**	NR	**+**	**+**	**+**	−	NR	**+**
[50]	NR	NR	**+**	NR	**+**	**+**	NR	**+**	NR	**+**	NR	NR	NR
[51]	NR	NR	**+**	−	−	**+**	−	−	−	−	NR	NR	NR
[52]	NR	NR	**+**	**+**	**+**	**+**	−	NR	−	NR	NR	NR	NR
[53] (w/PDD)	NR	NR	**+**	**+**	**+**	**+**	**+**	**+**	**+**	**+**	−	−	−
[53] (w/o PDD)	NR	NR	−	**+**	−	−	−	−	−	−	−	−	−
[54]	NR	NR	**+**	**+**	**+**	**+**	−	NR	NR	NR	NR	NR	NR
[55]	NR	NR	**+**	**+**	**+**	NR	**+**	NR	**+**	NR	**+**	**+**	NR
[56]	NR	NR	**+**	**+**	**+**	NR	**+**	**+**	NR	**+**	NR	NR	NR
[57]	NR	NR	**+**	−	**+**	**+**	**+**	−	**+**	**+**	−	NR	NR
[58]	NR	NR	**+**	**+**	**+**	NR	**+**	NR	NR	NR	NR	NR	NR
[59]	NR	NR	**+**	−	−	**+**	NR	**+**	NR	NR	NR	NR	**+**
[60]	NR	NR	**+**	**+**	NR	**+**	NR	**+**	NR	NR	NR	NR	NR
[61]	NR	NR	**+**	**+**	**+**	**+**	NR	NR	NR	NR	NR	NR	NR
**b.** Brain iron accumulation (BIA) in PD: present (+), absent (−), not reported (NR). Under each brain region is the number of studies reporting BIA as a fraction of the number of studies examining and reporting on this brain region.													
**Study number**	**Substantia Nigra**	**Red Nucleus**	**Parietal** **Lobe**	**Occipital Lobe**	**Frontal Lobe**	**Temporal Lobe**	**Cerebellum**	**Cingulate** **Gyrus**	**Thalamus**	**Caudate Nucleus**	**Putamen**	**Globus** **Pallidus**	**Hippocampus**
	13/15	6/14	1/1	0/1	2/2	2/2	0/0	0/0	3/10	1/13	3/15	5/14	2/2
[19]	**+**	NR	NR	NR	**+**	**+**	NR	NR	NR	NR	**+**	NR	NR
[62]	**+**	**+**	NR	NR	NR	NR	NR	NR	**+**	−	−	**+**	NR
[63]	**+**	**+**	NR	NR	NR	NR	NR	NR	NR	NR	**+**	**+**	NR
[64]	**+**	**+**	NR	NR	NR	NR	NR	NR	NR	−	−	−	NR
[65]	**+**	**+**	NR	NR	NR	NR	NR	NR	−	**+**	**+**	**+**	NR
[66]	**+**	**+**	NR	NR	NR	NR	NR	NR	−	−	−	**+**	NR
[67]	**−**	−	NR	NR	NR	NR	NR	NR	**+**	−	−	−	**+**
[68]	**+**	**+**	NR	NR	NR	NR	NR	NR	**+**	−	−	**+**	NR
[69]	**+**	−	**+**	−	**+**	**+**	NR	NR	−	−	−	−	**+**
[70]	**+**	−	NR	NR	NR	NR	NR	NR	−	−	−	−	NR
[71]	**+**	−	NR	NR	NR	NR	NR	NR	NR	−	−	−	NR
[72]	**+**	−	NR	NR	NR	NR	NR	NR	−	−	−	−	NR
[73]	**+**	−	NR	NR	NR	NR	NR	NR	−	−	−	−	NR
[74]	**+**	−	NR	NR	NR	NR	NR	NR	NR	−	−	−	NR
[75]	−	−	NR	NR	NR	NR	NR	NR	−	−	−	−	NR

**Table 3 pharmaceuticals-15-00551-t003:** Summary of the number of studies reporting GHM or BIA as a fraction of the number of studies examining and reporting on this brain region in AD and PD.

Brain Region	AD GHM	AD BIA	Brain Region	PD GHM	PD BIA
Temporal Lobe	10/10	5/5	Temporal Lobe	10/11	2/2
Parietal Lobe	9/9	2/2	Parietal Lobe	15/15	1/1
Frontal Lobe	3/3	1/3	Frontal Lobe	11/14	2/2
Occipital Lobe	2/3	0/0	Occipital Lobe	11/14	0/1
Hippocampus	3/3	5/5	Hippocampus	2/3	2/2
Cingulate Gyrus	9/9	1/1	Cingulate Gyrus	6/8	0/0
Caudate Nucleus	1/2	6/7	Caudate Nucleus	5/6	1/13
Putamen	0/1	6/6	Putamen	2/6	3/15
Globus Pallidus	0/0	2/6	Globus Pallidus	2/4	5/14
Substantia Nigra			Substantia Nigra	0/0	13/15
Red Nucleus			Red Nucleus	0/0	6/14
Precuneus	5/5	0/0			
Thalamus			Thalamus	5/8	3/10
Cerebellum			Cerebellum	6/10	0/0

## Data Availability

Data is contained within the article.

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
