# Peer review of "Brain Glucose Hypometabolism and Iron Accumulation in Different Brain Regions in Alzheimer’s and Parkinson’s Diseases"

_pharmaceuticals, 2022, doi:10.3390/ph15050551_

Round 1
Reviewer 1 Report
The authors face a very interesting problem that will certainly attract the interest of readers.
To improve the graphics and tables. The tables are the central part of the work, so they need to be explained better. Even the one figure needs improvement.
Author Response
We thank the reviewer for their helpful comments and suggestions. To improve the clarity, we first changed Tables 1a, 1b, 2a and 2b by summarizing the number of studies demonstrating the presence of GHM or BIA as a fraction of the total number of studies examining and reporting the biomarker in each brain region. We also changed the text in both the table legends and Results section to reflect these changes. We also explained that Figure 1 depicts the distribution of GHM and BIA in the brain based on the results summarized in Table 3.
Reviewer 2 Report
I have read the manuscript (ID: pharmaceuticals-1618538) that have been posted on 25 Feb 2022.
This manuscript by Indira Y. Rao and colleague summarized the brain regional distribution of glucose hypometabolism and iron accumulation in the patient with Alzheimer’s disease and Parkinson’s disease. The manuscript is well-written containing important information. The publication would be appropriate after minor revision.
A few minor points:
- In table 3 (also table 1a, 1b, 2a and 2b), the authors summarized the number of studies reporting GHM and BIA in each brain region. However, not all the listed studies have reported GHM or BIA. I think that showing GHM or BIA Absent (-) in the specific brain regions and Not Reported (NR) are big difference. I recommend adding the percentage, ratio, or some index of studies showing GHM or BIA Present (+) in all the listed studies excluding NR ones.
- In Figure 1, the brain regional distribution of GHM and BIA in the brain of patients with Alzheimer’s disease or Parkinson disease is colored in the drawing. However, I feel that the criterion for distribution of GHM and BIA is unclear. It would be better explaining its criterion in the legend of Figure 1 or somewhere else.
Author Response
We thank the reviewer for their helpful comments and suggestions.
1) To improve the clarity, we first changed Tables 1a, 1b, 2a and 2b, according to your suggestion, by summarizing the number of studies demonstrating the presence of GHM or BIA as a fraction of the total number of studies examining and reporting the biomarker in each brain region. We also changed the text in both the table legends and Results section to reflect these changes.
2) We also explained that Figure 1 depicts the distribution of GHM and BIA in the brain based on the results summarized in Table 3.
Reviewer 3 Report
This is a brief critical review which address an important issue in Alzheimer’s and Parkinson’s diseases. The main points of this manuscript are well identified and the authors reach a significant conclusion based on the review of selected data published and selected using criteria that are clearly defined in the manuscript. The analysis of the published data has been carefully performed. However, a more in-depth analysis of the correlations between GHM and BIA with tau-PET imaging data and amyloid β deposition in the brain of Alzheimer’s disease (AD) will be desirable to further highlight the relative relevance of cross-correlations between GHM and BIA with tau and amyloid β deposition. This could be achieved including an additional Figure similar to the Figure 1, or more panels in this Figure 1. It is recommended to add these data to largely improve and strengthen the main conclusions of this work.
Other points:
-Ref. 19 seems to be incomplete. Is it a web page or an article in a scientific journal?
-Ref. 30 seems to be incomplete. The authors are lacking in this reference. Why an abstract is cited for this important point instead of a more complete paper published in an indexed scientific journal?
Author Response
We thank the reviewer for their helpful comments and suggestions. We have corrected the citations of the two references identified by the reviewer. The first reference is to a U.S. patent, and the second was to an article in an indexed scientific journal. We carefully considered the suggestion to add a figure and/or text showing correlations between GHM and BIA with tau-PET imaging and amyloid β deposition in Alzheimer’s disease. As the focus of this review is to highlight GHM and BIA as under-appreciated biomarkers and potential alternative therapeutic targets, we are concerned that making the suggested addition will detract from this. Further, as stated in our manuscript, we feel that “Hypothesis-driven FDG-PET and QSM-MRI imaging studies, where both are conducted on each group of individuals with AD or PD, are needed to confirm or disprove the tentative observations presented here about the potential relationship or lack thereof between GHM and BIA in AD and PD.” Once this has been done, we would be more comfortable correlating those findings with tau-PET imaging and amyloid β deposition in Alzheimer’s disease.
Reviewer 4 Report
The paper reviews literature reporting investigations by imaging techniques in different brain regions with focus on brain glucose hypometabolism and brain iron accumulation. The selection of the reviewed papers is sound.
Minor comments:
Introduction, Line 69: defective hopmeostasis of iron: please better refer to the impaired homeostasis of the redox balance of iron species Fe2+ vs. Fe3+ since mainly this is the cause of derailed oxidative stress.
Line 75: Here you refer to post mortem brains and iron: specifically the iron redox balance in post mortem tissue cannot reflect any situation ind living systems. further, later in your paper in Methods section you (correctly) exclude post mortem studies from your reviewed papers.
Discussion:
You refer to AD and PD brains and their regions. It should be mede more clear in discussion that observed accumulation or changes were related to controls in the original studies. The amount of changes compared to controls should be more clearly ruled out.
Author Response
We thank the reviewer for their helpful comments and suggestions.
- As recommended, we added the following to the Introduction, “The impaired redox balance of the iron species Fe2+ vs. Fe3+ contributes to oxidative stress [9].”
- As indicated by the reviewer, we state in the Methods that “Postmortem studies were excluded.” In accordance with this, we did not include in the Tables any data resulting from postmortem studies as we agree this would not be appropriate. Many well-designed preclinical studies of postmortem brain tissue have contributed to our understanding of brain disorders.
- We agree that it needs to be stated that the results shown in the Tables and Figure represent regions where GHM and BIA were elevated when compared to healthy controls. In our Methods section, it now states: “Studies that conducted FDG-PET scans to locate regions with elevated GHM in the brains of patients with AD or PD as compared to healthy controls were examined. Also, studies that conducted QSM-MRI on AD or PD as compared to healthy controls brains were selected and regions where these studies reported elevated BIA were examined”
The first sentence of the Discussion section now states: “This is the first study to explicitly describe the spatial relationship in the brain between GHM and BIA, two biomarkers that are elevated in comparison to healthy controls in both AD and PD, by comparing data from brain FDG-PET and QSM-MRI studies.” While the multiple published papers analyzed in this review provide proof of statistically significant elevations of the biomarker they studied, they do not all state the exact quantitative differences in the amount of GHM or BIA present in comparison to healthy controls. Therefore, it is not possible to discuss the amount of changes compared to controls in this review.
Round 2
Reviewer 3 Report
I have no further comments.